# Calibration of isotopologue-specific optical trace gas analysers: A practical guide

David W. T. Griffith[1],

[1] Centre for Atmospheric Chemistry, University of Wollongong, Australia

*Correspondence to*: David Griffith (Griffith@uow.edu.au)

**Abstract**

The isotopic composition of atmospheric trace gases such as $CO_2$ and $CH_4$ provides a valuable tracer for the sources and sinks that contribute to atmospheric trace gas budgets. In the past, isotopic composition has typically been measured with high

precision and accuracy by Isotope Ratio Mass Spectrometry (IRMS) offline and separately from real-time or flask-based measurements of concentrations or mole fractions. In recent years, development of infrared optical spectroscopic techniques based on laser and Fourier Transform Infrared spectroscopy has provided high precision measurements of the concentrations of one or more individual isotopologues of atmospheric trace gas species in continuous field and laboratory measurements, thus providing both concentration and isotopic measurements simultaneously. Several approaches have been taken to the

calibration of optical isotopologue-specific analysers to derive both total trace gas amounts and isotopic ratios, converging into two different approaches: calibration via the individual isotopologues as measured by the optical device, and calibration via isotope ratios, analogous to IRMS.

This paper sets out a practical guide to the calculations required to perform calibrations of isotopologue-specific optical

analysers, applicable to both laser or broadband FTIR spectroscopy. Equations to calculate the relevant isotopic and total concentration quantities without approximation are presented, together with worked numerical examples from actual measurements. Potential systematic errors which may occur when all required isotopic information is not available, or is approximated, are assessed. Fortunately, in most such realistic cases, these systematic errors incurred are acceptably small and within the compatibility limits specified by the World Meteorological Organisation – Global Atmosphere Watch.

Isotopologue-based and ratio-based calibration schemes are compared. Calibration based on individual isotopologues is simpler because the analysers fundamentally measure amounts of individual isotopologues, not ratios. Isotopologue calibration does not require a range of isotopic ratios in the reference standards used for the calibration, only a range of concentrations or mole fractions covering the target range. Ratio-based calibration leads to concentration dependence which must also be characterised.

# 1 Introduction

Until recently, measurements of the amounts of $CO_2$ and other trace gases in the atmosphere and in calibration gas standards within the Global Atmosphere Watch - Greenhouse Gas Monitoring Techniques (GAW-GGMT) community were mostly made by analytical techniques which do not discriminate between isotopic variants of the target gases. Manometry and gravimetry enable the calibration of gas mixtures to be traceable to SI units of pressure, volume, mass and temperature, but measure only the total amounts of the target trace gas, without taking into account differences in isotopic composition. Gas chromatography is also commonly used both in atmospheric measurements and in the propagation of standards, but is also blind to the isotopic composition of the target gas and measures only total amounts.

Non-dispersive infrared analysers (NDIRs) have been used for many years as an instrument of choice for atmospheric trace gas monitoring. NDIR is an optical technique based on infrared absorption by the target trace gas, and like any optical/spectroscopic instrument, NDIR instruments have a different response to different isotopologues of the target species because different isotopologues have different absorption spectra. Earlier NDIR instruments such as URAS, UNOR, Siemens and APC employed microphone detectors filled with the target trace gas that responded selectively to the absorption of infrared radiation by the target gas in the sample (Griffith, 1982). The NDIR instrument response depends in a complex and non-linear way on the isotopic composition of the target gas and on the carrier gas. The more recent Licor instruments replaced the microphonic detector with an optical semiconductor detector which relies on a broad bandpass filter to restrict the wavelength range from the source to that absorbed by the target gas, for example for $CO_2$ around 4.3 μm. Optical NDIR detectors also respond differently to the different isotopologues of the target gas because the bandpass filter does not cover the entire absorption range of the trace gas, and because different isotopologues have different absorption strengths and sensitivities. NDIR instruments thus have an ill-defined sensitivity to isotopic variability which must be empirically quantified for the most precise atmospheric measurements (Lee et al., 2006; Tohjima et al., 2009).

Most recently, laser and Fourier Transform Infrared (FTIR) based optical infrared analysers have taken on a major role in atmospheric trace gas measurements for many gases, especially the dominant greenhouse gases $CO_2$ and $CH_4$. These instruments are based on infrared absorption by single absorption lines or bands of specific isotopologues, which are only a proxy for the total amount of the target trace gas. If the isotopic composition of the trace gas is invariant, such analysis provides a valid measure of the total amount of the gas after calibration, but it has long been recognised that isotopic differences between the calibration gases and the samples measured lead to variations in the total trace gas amounts deduced from a single isotopologue measurement that are significant relative to GAW compatibility goals (Loh et al., 2011). Several studies have addressed isotopic calibration (e.g. Esler et al., 2000; Bowling et al., 2003; Griffis et al., 2005; Mohn et al., 2008; Loh et al., 2011; Tuzson et al., 2011; Griffith et al., 2012; Wehr et al., 2013; Wen et al., 2013; Rella et al., 2015; Vardag et al., 2015; Pang et al., 2016; Flores et al., 2017; Tans et al., 2017; Braden-Behrens et al., 2017) and compared calibration approaches (Wen et al., 2013), but until recently most studies made some level of approximation in dealing with the calculations required to properly include the contributions of all possible isotopologues of the target species in the calculation scheme. Most recently Griffith et al (2012), Flores et al. (2017) and Tans et al. (2017) have published isotopic calibration strategies which are equivalent and which correctly and completely account for the full isotopic composition of the target gas ($CO_2$ in these studies, but applicable in principle to any species).

Established calibration laboratories using mass spectrometry as the primary method for isotopic analysis normally provide calibration standards which specify total amount and isotopic ratios of a trace gases in an air matrix, such as $CO_2$, $\delta^{13}C$ and $\delta^{18}O$, while optical analysers fundamentally determine individual isotopologue amounts of isotopologues such as $^{16}O^{12}C^{16}O$, $^{16}O^{13}C^{16}O$ and $^{16}O^{12}C^{18}O$. Here we present a practical guide to the calculations required to rigorously but simply convert

between the two equivalent descriptions and to derive isotope-specific calibrations for optical analysers. The calculations described here are equivalent to those described by Wehr et al. (2013), Flores et al. (2017) and Tans et al. (2017). The motivation for this technical note is thus threefold:

- to show that the complete and correct treatment of isotopic composition in calibration calculations is straightforward and that there is no need to invoke some approximations often made in earlier analyses,
- to provide a practical guide to isotope-specific calibration calculations, and
- to assess the potential errors when all isotopic information is not available and approximations or assumptions must be made.

## 2    Calculation of isotopic quantities

Using $CO_2$ as an example, considering the stable isotopes $^{12}C$, $^{13}C$, $^{16}O$, $^{17}O$ and $^{18}O$, there are eighteen possible isotopologues (2 x 3 x 3 isotopic possibilities). $^{14}C$ is a negligible proportion of total carbon for these purposes and is neglected. Only twelve of these eighteen possibilities are distinct due to symmetry. Assuming the substitution of each isotope at each position in the molecule follows its bulk statistical abundance (i.e. no clumping, see section 6), only four independent quantities are required to fully define the total amount and full isotopic composition of $CO_2$. These quantities may equivalently be the total $CO_2$ amount and three isotopic ratios $^{13}r$, $^{17}r$ and $^{18}r$ (or delta values $\delta^{13}C$, $\delta^{17}O$ and $\delta^{18}O$), or the amounts of four individual isotopologues with each isotope substituted, most conveniently $^{16}O^{12}C^{16}O$, $^{16}O^{13}C^{16}O$, $^{16}O^{12}C^{17}O$ and $^{16}O^{12}C^{18}O$. Once these are known, the abundances of all multiply-substituted isotopologues can be calculated.

The most fundamental quantity defining isotopic composition for each element is the *isotope ratio* of the minor to the major isotope

$$^{13}r = \frac{n(^{13}C)}{n(^{12}C)}$$

$$^{17}r = \frac{n(^{17}O)}{n(^{16}O)} \tag{1}$$

$$^{18}r = \frac{n(^{18}O)}{n(^{16}O)}$$

where for example $n(^{13}C)$ is the amount of $^{13}C$ in a sample (number of moles or atoms). Isotope ratios for standard or reference materials are assigned by the isotope metrology community, (e.g. Allison et al., 1995; Brand et al., 2010; Werner and Brand, 2001).

Isotope ratios are commonly expressed as delta values relative to a standard or reference material

$$\delta^{13}C = \left( \frac{^{13}r}{^{13}r_{ref}} - 1 \right)$$

$$\delta^{17}O = \left( \frac{^{17}r}{^{17}r_{ref}} - 1 \right) \tag{2}$$

$$\delta^{18}O = \left( \frac{^{18}r}{^{18}r_{ref}} - 1 \right)$$

(Following the recommendation of Coplen (2011) and to simplify equations, the factor 1000 ‰ is not included in the definition of δ.) For the relevant reference scales commonly used in atmospheric analysis, the reference isotope ratios are given in Table 1.

For each isotope of an element, the *isotopic abundance* or *isotopic fraction* is the fraction of that isotope relative to all isotopes in a sample

$$^{12}x = \frac{n(^{12}C)}{n(^{12}C)+n(^{13}C)} = \frac{1}{(1+{}^{13}r)}$$

$$^{13}x = \frac{n(^{13}C)}{n(^{12}C)+n(^{13}C)} = \frac{{}^{13}r}{(1+{}^{13}r)}$$

$$^{16}x = \frac{n(^{16}O)}{n(^{16}O)+n(^{17}O)+n(^{18}O)} = \frac{1}{(1+{}^{17}r+{}^{18}r)}$$

$$^{17}x = \frac{n(^{17}O)}{n(^{16}O)+n(^{17}O)+n(^{18}O)} = \frac{{}^{17}r}{(1+{}^{17}r+{}^{18}r)}$$

$$^{18}x = \frac{n(^{18}O)}{n(^{16}O)+n(^{17}O)+n(^{18}O)} = \frac{{}^{18}r}{(1+{}^{17}r+{}^{18}r)}$$

(3)

Note that these are fractional abundances such that $^{12}x + {}^{13}x = 1$ and $^{16}x + {}^{17}x + {}^{18}x = 1$.

Similarly, for a molecule the *isotopologue abundances* or *isotopologue fractions* are defined – for example for $CO_2$ the isotopologue abundances for $^{12}C^{16}O_2$ (626), $^{13}C^{16}O_2$ (636), $^{12}C^{16}O^{18}O$ (628) and $^{12}C^{16}O^{17}O$ (627) are:

$$x_{626} = {}^{16}x \cdot {}^{12}x \cdot {}^{16}x = \frac{1}{R_{sum}}$$

$$x_{636} = {}^{16}x \cdot {}^{13}x \cdot {}^{16}x = \frac{{}^{13}r}{R_{sum}}$$

$$x_{627} = 2 \cdot {}^{16}x \cdot {}^{12}x \cdot {}^{17}x = \frac{2 \cdot {}^{17}r}{R_{sum}}$$

$$x_{628} = 2 \cdot {}^{16}x \cdot {}^{12}x \cdot {}^{18}x = \frac{2 \cdot {}^{18}r}{R_{sum}}$$

(4)

where

$$R_{sum} = (1+{}^{13}r) \cdot (1+{}^{17}r+{}^{18}r)^2$$

(5)

The labels 626, 636, 628, 627 are the common isotopic shorthand used in spectroscopy and the Hitran database. The sum of all isotopologue abundances $x$ over all 18 isotopologues is equal to unity. $R_{sum}$ is a sum of 18 products of isotope ratios, one corresponding to each of the 18 possible isotopologues of $CO_2$. $R_{sum}$ conveniently accounts for all possible isotopologues in calculations of abundances, providing a normalising factor somewhat analogous to a partition sum over all energy levels of a molecule. From Eq. (4), $x_{626} = 1/R_{sum}$ i.e. $1/R_{sum}$ is the fractional abundance of the major isotopologue and $R_{sum} - 1 \approx 1 - x_{626}$ is that fraction of the sample that is made up of all the minor isotopologues. Equivalently, from Eq. (10) and the following paragraph it can be seen that $R_{sum}$ is the ratio of the total amount of $CO_2$ to that of the major isotopologue in a sample.

Abundances of the major and three singly-substituted isotopologues and $R_{sum}$ values for standard reference scales are given in Table 2. Abundances of the multiply-substituted isotopologues can be calculated following the examples of Eq. (4). They are also listed for Hitran isotope ratios on the Hitran website https://www.cfa.harvard.edu/hitran/molecules.html.

For a calibration or reference gas, $\delta^{13}C$ and $\delta^{18}O$ are usually provided by calibration laboratories, and $\delta^{17}O$ can normally be deduced from $\delta^{18}O$ assuming mass dependent fractionation of oxygen isotopes with negligible error (Brand et al., 2010):

$$^{17}r / {}^{17}r_{ref} = ({}^{18}r / {}^{18}r_{ref})^{0.528}$$

$$or \tag{6}$$

$$\delta^{17}O = 0.528 \cdot \delta^{18}O$$

The mass dependent fractionation assumption is discussed below in section 6. The isotope ratios $^{13}r,$ $^{17}r$ and $^{18}r$ for a sample

can be thus be calculated from inverting equations (2)

$$^{13}r = (1 + \delta^{13}C) \cdot {}^{13}r_{ref}$$

$$^{17}r = (1 + \delta^{17}O) \cdot {}^{17}r_{ref} \tag{7}$$

$$^{18}r = (1 + \delta^{18}O) \cdot {}^{18}r_{ref}$$

thence $R_{sum}$ can be calculated from (5) for any sample or reference gas.

If the total mole fraction of $CO_2$ in a sample of air, $y_{CO2}$, is also known (for example, for a certified calibration gas), the

individual isotopologue amounts or mole fractions can be calculated from

$$y_{626} = y_{CO2} \cdot x_{626} = y_{CO2} / R_{sum}$$

$$y_{636} = y_{CO2} \cdot x_{636} = y_{CO2} \cdot {}^{13}r / R_{sum}$$

$$y_{627} = y_{CO2} \cdot x_{627} = y_{CO2} \cdot 2 \cdot {}^{17}r / R_{sum} \tag{8}$$

$$y_{628} = y_{CO2} \cdot x_{628} = y_{CO2} \cdot 2 \cdot {}^{18}r / R_{sum}$$

(Following the recommendation of the IUPAC Gold book (McNaught and Wilkinson, 2014) and usage by Tans et al. (2017), the symbol $y$ is used here for mole fraction (more formally amount fraction) of a trace gas or isotopologue in air, to distinguish from $x$, the isotope or isotopologue fractional abundance.)

Conversely, if a set of calibrated isotopologue mole fractions $\{y_{626}, y_{636}, y_{628}, y_{627}\}$ in a sample are measured with an isotopologue-specific analyser, the total $CO_2$ mole fraction $y_{CO2}$ and isotope ratios or delta values can be calculated. The isotope ratios are derived directly from the isotopologue amounts,:

$$^{13}r = y_{636} / y_{626}$$

$$^{18}r = 0.5 \cdot y_{628} / y_{626} \tag{9}$$

$$^{17}r = ({}^{18}r / {}^{18}r_{ref})^{0.528} \cdot {}^{17}r_{ref}$$

then delta values are calculated from (2) and $R_{sum}$ from (5). The total $CO_2$ mole fraction is then calculated from (8):

$$y_{CO2} = y_{626} \cdot R_{sum} \tag{10}$$

The key quantity in these calculations is $R_{sum}$, which correctly and completely accounts for all possible isotopologues of the molecule at their actual isotopic abundances. Note that to correctly calculate the amount of *any* isotopologue in a sample, *all* isotope ratios should be known to calculate $R_{sum}$ exactly. Errors incurred when this requirement is relaxed are discussed and quantified in section 5.

## 3 Normalised isotopologue mole fractions

In the Hitran database, tabulated line strengths are normalised by the natural abundance of the relevant isotopologue; the reference isotopologue natural abundances assumed in Hitran are listed in Table 2. Retrievals from spectra based on Hitran line parameters thus provide scaled or normalised mole fractions of isotopologues which are referenced to the isotopic scales assumed by Hitran. For some purposes it may be convenient to work with these normalised mole fractions directly rather than to convert them to absolute mole fractions as in section 2 because the reference isotopologue abundances are inherently included in the normalised amounts. In terms of normalised mole fractions, equations (8) become:

$$y'_{626} = \frac{y_{626}}{x_{626,ref}} = y_{CO2} \cdot \frac{R_{sum,ref}}{R_{sum}} = y_{CO2} / X_{sum}$$

$$y'_{636} = \frac{y_{636}}{x_{636,ref}} = y_{CO2} \cdot \frac{^{13}r}{^{13}r_{ref}} \cdot \frac{R_{sum,ref}}{R_{sum}} = y_{CO2} \cdot (1+\delta^{13}C) / X_{sum}$$

$$y'_{627} = \frac{y_{627}}{x_{627,ref}} = y_{CO2} \cdot \frac{^{17}r}{^{17}r_{ref}} \cdot \frac{R_{sum,ref}}{R_{sum}} = y_{CO2} \cdot (1+\delta^{17}O) / X_{sum} \tag{11}$$

$$y'_{628} = \frac{y_{628}}{x_{628,ref}} = y_{CO2} \cdot \frac{^{18}r}{^{18}r_{ref}} \cdot \frac{R_{sum,ref}}{R_{sum}} = y_{CO2} \cdot (1+\delta^{18}O) / X_{sum}$$

where $r_{ref}$ and $R_{sum,ref}$ refer to the reference scales listed in Table 1 and Table 2 and $X_{sum} = R_{sum} / R_{sum,ref} = R_{sum} \cdot x_{626,ref}$ .

Equations (11) allow normalised mole fractions to be calculated from total $CO_2$ mole fraction and $\delta$ values on any reference scale for which $r_{ref}$ and $R_{sum,ref}$ are known.

Calculation of $\delta$ values from normalised isotopologue mole fractions is analogous to Eq. (9) and (10):

$$\delta^{13}C = \frac{y'_{636}}{y'_{626}} - 1$$

$$\delta^{18}O = \frac{y'_{628}}{y'_{626}} - 1 \tag{12}$$

$$\delta^{17}O = 0.528 \cdot \delta^{18}O$$

and the total $CO_2$ mole fraction is

$$y_{CO2} = y'_{626} \cdot \frac{R_{sum}}{R_{sum,ref}} = y'_{626} \cdot X_{sum} \tag{13}$$

The normalised mole fractions have the convenient property that they are all equal to the total $CO_2$ mole fraction in a sample if all isotopes are in natural abundance in the reference scale (i.e. Eq. (11) with $\delta = 0$, $R_{sum} = R_{sum,ref}$ and $X_{sum} = 1$). Hitran natural abundances are based on a superseded definition of VPDB isotope ratio for carbon and SMOW for oxygen, while for atmospheric $CO_2$ the isotopic scale of choice is VPDB-$CO_2$, which is based on VPDB for both carbon and oxygen, and may be adjusted over time as scales are re-determined. To convert normalised mole fractions retrieved directly from spectra (Hitran scale) to the VPDB-$CO_2$ scale, each normalised mole fraction can be multiplied by $x_{ref,Hitran} / x_{ref,VPDB}$. The reference isotopologue abundances and rescaling factors are listed in Table 2.

## 4 Calibration and measurement procedures – step by step

Calibration of an isotopologue-specific analyser can in principle be carried out in two ways, calibrating on either the individual isotopologue amounts or on the derived isotope ratios or delta values. Both methods have been used in the published work to

date. The former is more fundamental because optical methods actually measure individual isotopologue amounts, not ratios. Ratio or delta-based calibration leads to the additional complication of concentration dependence in the calibration. A step by step method for direct isotopologue calibration is presented in section 4.1 based on the equations of section 2, ratio or delta calibration is discussed in section 4.2, and the two methods are compared in section 4.3.

## 4.1 Direct calibration by isotopologue amounts

The steps described here are consistent with those recently-published by Flores et al. (2017) and Tans et al. (2017). Griffith et al. (2012) previously described the same methods but used a minor approximation in accounting for the sum of all multiply-substituted isotopologues in the calculation of $R_{sum}$ in Eq (5) or $X_{sum}$ in Eq (11).

There are two parts to the calibration and unknown measurement procedure: (1) determination of the reference isotopologue amounts and the calibration equation for each isotopologue in a calibration gas, and (2) measurement of the isotopologue amounts in an unknown sample and calculation of its total trace gas amount and delta quantities. As above $CO_2$ is used as an example, but the procedures apply in principle to any molecule.

*Calibration*

1.  From reference standard tank data provided by the calibration laboratory $\{CO_2, \delta^{13}C, \delta^{18}O, (\delta^{17}O)\}$, calculate isotope ratios $^{13}r, ^{18}r, ^{17}r$ and $R_{sum}$ for each standard (Eq. (7) then Eq. (5)).

2.  Calculate the calibrated amount of each isotopologue $y_{626}, y_{636}, y_{628}$ in each standard (Eq (8)).

3.  Measure uncalibrated analyser responses or raw isotopologue amounts of each standard $y_{626,meas}, y_{636,meas}, y_{628,meas}$ with the analyser.

4.  Derive the calibration equation for each isotopologue, for example for a linear calibration

$$y_{626,meas} = a_{626} \cdot y_{626} + b_{626} \tag{14}$$

*Sample measurement*

1.  Measure the sample with the analyser and determine the analyser responses or raw isotopologue amounts.

2.  Apply the inverted calibration determined in 4. above for each isotopologue to determine calibrated isotopologue amounts.

3.  Calculate $^{13}r, ^{18}r, ^{17}r$ and $R_{sum}$ from calibrated isotopologue amounts (Eq. (9))

4.  Calculate $\delta^{13}C$ and $\delta^{18}O$ on the desired reference isotope scale (Eq. (2) or (12)).

5.  Calculate total $CO_2$

$$y_{CO2} = y_{626} \cdot R_{sum} \qquad \text{(Eq. (10))}$$

With this scheme, for complete calibration of the analyser the total $CO_2$ amount, $\delta^{13}C$ and $\delta^{18}O$ should be known for each reference standard, and each isotopologue should be measured by the analyser (or a combination of analysers). $\delta^{17}O$ can be calculated with sufficient accuracy from $\delta^{18}O$. Calibration gases may but do not need to span a range of delta values, they need only span the range of amounts of each isotopologue covered by the range of samples to be measured(Bowling et al., 2003). Flores et al. (2017) demonstrated isotopic calibration of $CO_2$ in which all standards were synthesised from the same $CO_2$ source gas and all had the same $\delta^{13}C$ and $\delta^{18}O$ values.

## 4.2    Calibration by delta values

Spectroscopic analysers fundamentally determine the amounts of individual isotopologues, and the isotopologue-based analysis as described in the preceding section is the natural choice as a basis for calibration. Historically however, isotope ratio mass spectrometry (IRMS) has been the method of choice for isotopic analysis because many sources of noise cancel in calculating the ratio. Traditional IRMS calibration schemes are based on standards over a range of isotope ratios or delta values directly rather than on isotopologue amounts. Ratio or delta calibration schemes have thus, perhaps inevitably, flowed through to optical techniques. Ratio calibration schemes use calibration standards which cover a range of delta values and derive calibration equations analogous to Eq. (14) directly in terms of delta values rather than isotopologue amounts. The raw measured delta values are calculated from the uncalibrated isotopologue amounts. However, as shown in the following, this method inevitably leads to a concentration dependence of the calibration equations which must be characterised as part of (and that significantly complicates) the calibration procedure.

Several groups have reported on ratio calibration schemes and the consequent concentration dependence (e.g. Griffith et al., 2012; Wen et al., 2013; Rella et al., 2015; Pang et al., 2016; Braden-Behrens et al., 2017; Flores et al., 2017). The concentration dependence inevitably follows if the actual calibration relationships between measured and true amounts of individual isotopologues (section 4.1, Eq. (14)) have a non-zero y-intercept or an additional non-linear term. Griffith et al. (2012, Eq. 14) showed that a non-zero intercept in the calibration equations leads to an approximate inverse concentration dependence of measured $\delta^{13}C$. Extending that to include a quadratic term in the calibration equation representing non-linearity adds an approximately linear term to the concentration dependence, which can then be described by a combination of an inverse and linear dependence on $y_{CO2}$:

$$\delta^{13}C_{meas} = \alpha \cdot \delta^{13}C_{true} + (\alpha - 1) + \frac{\beta}{y_{CO2}} + \gamma \cdot y_{CO2} \tag{15}$$

where $\delta^{13}C_{meas}$ is calculated from the *raw* measured isotopologue amounts. For a perfectly linear calibration equation (14) with $b_{626} = b_{636} = 0$ both $\beta$ and $\gamma$ are zero, $\alpha = a_{636} / a_{626}$ and Eq. (15) represents a simple concentration-independent scale shift of $(\alpha-1)$ in the $\delta$ scale. $\beta$ is a function of the intercept terms $b_{626}$ and $b_{636}$, and $\gamma$ becomes non-zero if non-zero quadratic terms are added to the calibration equations. The inverse and linear $y_{CO2}$ dependences are not exact because the coefficients $\beta$ and $\gamma$ contain terms dependent on $\delta^{13}C$ and there are weak cross-terms, but together they provide a useful model to describe the concentration dependence. The linear term becomes relatively more important than the inverse term at high $CO_2$ mole fractions where the inverse $CO_2$ term becomes small and any quadratic contribution to the calibration equation leading to the linear term becomes large.

Figure 1 illustrates this concentration dependence with a typical $\delta^{13}C$ vs $CO_2$ dependence for an FTIR analyser similar to that used in the example of section 5 below. The dependence was determined by continuous flow measurements of a single $CO_2$-spiked air tank while the $CO_2$ content was gradually reduced by passing a fraction of the flow through Ascarite. The measured $\delta^{13}C$ vs $CO_2$ data are fitted to Eq. (15) with fitted parameters $\beta = - 1227$ ‰ ppm and $\gamma = 0.0054$ ‰ ppm$^{-1}$, corresponding to $CO_2$ - dependent corrections of up to 5‰ over the $CO_2$ range 400-1000 ppm. The residuals of the fit illustrate potential errors from the modelled behaviour of up to $\pm$ 0.3‰. Uncertainties in calibrating the $CO_2$ concentration dependence can lead to significant errors in Keeling-type analyses over a wide range of total $CO_2$ amounts even if the isotopologue calibration non-linearity is very small (Pang et al., 2016; Wen et al., 2013).

The concentration dependence is a function of the isotopologue calibration coefficients, and thus in principle for best accuracy it should be re-determined for every calibration, complicating the calibration procedure. The Thermo-Fisher Delta Ray isotope analyser, for example, takes this approach in a prescribed sequence of measurements using several reference standards;

however Braden-Behrens et al. (2017) and Flores et al. (2017) found this procedure not to be sufficiently accurate or stable and invoked separate a posteriori calibration schemes. Rella et al. (2015; Picarro, 2017) similarly describe a calibration procedure for Picarro analysers to take concentration dependence into account.

### 4.3 Comments on the accuracy of optical isotopologue and ratio calibration

As an example assume a calibration laboratory provides calibrated reference gases with an absolute accuracy of 0.05 ppm for total $CO_2$ amount (0.12‰ in 400 ppm $CO_2$) and 0.02‰ for $\delta^{13}C$ measured by IRMS. The isotope ratio is thus more accurately determined than the total amount fraction for the reference gases. Now take as a practical measurement repeatability for optical analysers 0.02 ppm (0.05‰) for total $CO_2$ amount and 0.07‰ for $\delta^{13}C$ (e.g. Griffith et al. (2012), laser instruments are similar). The absolute accuracy for the calibrated optical measurement of total $CO_2$ is limited by the reference gas amount fraction, but the more accurately known reference $^{13}r$ or 626/636 ratio is carried through the calibration calculations and this accuracy is preserved when retrieved isotopologue amounts are ratioed. The accuracy of measured $^{13}r$ or $\delta^{13}C$ is thus limited by the optical measurement (0.07‰) which is less precise than the IRMS-provided reference accuracy (0.02‰). This reasoning applies to both isotopologue and ratio calibration schemes, which both benefit from the higher accuracy and precision in the isotopologue ratios than in absolute isotopologue amounts. The principle differences between the isotopologue and ratio calibration schemes are twofold:

- The isotopologue scheme does not require calibration gases spanning a range of delta values, it is sufficient to span the range of total amount fractions of interest. This simplifies the preparation of reference gases for calibration laboratories.

- The ratio scheme has an unavoidable $CO_2$ concentration dependence which must be characterised and leads potentially to a loss of accuracy, as shown in section 4.2. This complicates the calibration procedure for optical analysers.

Optical FTIR and laser methods do not currently meet GAW requirements for repeatability of $\delta^{13}C$ in $CO_2$ in clean background air measurements of 0.01‰ (WMO-GAW, 2016). Their precision is limited by the inherent signal:noise ratio of the optical measurement, not by the choice of absolute or ratio calibration. The precision currently available from optical measurements is nevertheless very useful for continuous analysis of air in non-baseline scenarios such as urban air or agricultural flux measurements.

Errors are discussed further in section 6.

### 5 Tutorial: a practical worked example

This section presents a worked example of the calibration of an optical analyser using reference gases of given total $CO_2$ mole fraction, $\delta^{13}C$ and $\delta^{18}O$, followed by measurements of air to which this calibration is applied. The data are derived from an Ecotech Spectronus FTIR analyser which measures three isotopologues of $CO_2$ (626, 636, 628) in the calibration gases and in the sampled air. The calculations follow section 4.1.

*Calibration*

The calibration data were collected in the laboratory at the University of Wollongong on 27 Sept 2017. Four reference tanks were sourced from CSIRO GASLAB, with total $CO_2$ mole fraction, $\delta^{13}C$ and $\delta^{18}O$ provided on the current WMO reference scales (WMO X2007 scale for total $CO_2$, VPDB-$CO_2$ for $\delta^{13}C$ and $\delta^{18}O$). For each calibration tank, $^{13}r$, $^{18}r$, $^{17}r$, $R_{sum}$, and reference isotopologue mole fractions are calculated from equations (7), (5) and (8). The four reference gases were measured

in the analyser, and raw measured values of the isotopologue mole fractions corrected to dry air and for small spectroscopic cross-sensitivities to pressure, temperature and water vapour as described by Griffith et al. (2012). A two-parameter linear regression (slope and intercept) of measured against reference mole fractions for each isotopologue provides the linear calibration coefficients $a$ and $b$ for the analyser, Eq. (14). The worked data are presented in Table 3 and calibration plots shown in Figure 2.

*Sample air measurements*

Figure 3 shows an example of one day of calibrated 1-minute measurements from the same FTIR analyser collected at a rural site in SE Australia on 23 and 24 Jan 2018. Table 4 illustrates the worked calibration of the raw data at four times of differing $CO_2$ amounts and isotopic fractionations. The linear calibration of 27 Sept 2017 described above has been applied to the measured data without further correction. The calculations follow section 4.1 to determine $y_{CO2}$, $\delta^{13}C$ and $\delta^{18}O$ for each 1-minute measurement. Figure 4 shows an example of a Keeling plot derived from the data of Figure 3, with an intercept -24.5‰ typical of the dominant plants in this agricultural area.

## 6    Assessment of potential errors

Table 5 shows examples of actual isotopologue amounts for samples with total $CO_2$ = 400 ppm and a range of isotopic compositions. The table includes $R_{sum}$ values calculated for each sample. The potential error incurred in calculating the total $CO_2$ amount from a spectroscopic measurement of $y_{626}$ via Eq. (10) if the different isotopic composition between sample and reference gases is not taken into account is shown in the rightmost column – it is the difference from 400 ppm of the total $CO_2$ calculated from Eq. (10) taking the reference value $R_{sum,ref}$ (case 1) instead of the correct value on the same line $R_{sum}$. This simulates the effect of ignoring the difference in isotopic composition between reference and sample. The reference case (case 1) is a hypothetical standard with the isotopic composition of VPDB-$CO_2$. Examples include typical clean air (case 2), synthetic air synthesised with $^{13}C$-depleted $CO_2$ with $\delta^{13}C$ = -35‰ (case 3), systematic errors of 2‰ in $\delta^{18}O$ and $\delta^{17}O$ (cases 4,5), and using isotope ratios assumed by Hitran rather than VPDB-$CO_2$ (case 6). Case 7 simulates the result if only singly-substituted isotopologues are included in the sum and all doubly-substituted minor isotopologues are ignored. Other cases can be assessed following the equations of section 2. Potential errors are fortunately small relative to GAW compatibility goals for realistic isotopic variations of a few per mil around clean air values. However the potential for significant errors (> 0.1 ppm) exists for reference gas mixtures or samples with $^{13}C$-depleted $CO_2$ as is often the case for synthetic mixtures or for samples with added $CO_2$ derived from plant or fossil fuel sources.

These potential errors in computation of delta values should also be viewed in the context of experimental measurement errors. Flores et al. (2017) formally evaluated the uncertainty budget for their particular FTIR measurements of $\delta^{13}C$ in $CO_2$ and found a standard uncertainty of 0.09‰, of comparable magnitude to the largest potential computational approximation errors. The measurement uncertainty was dominated by uncertainty in assigned reference mole fractions for the reference standards rather than the spectroscopic measurement uncertainty.

Three assumptions, previously mentioned and summarised here, have negligible impact on the calculations of section 2 and Table 5:
   -    $^{14}C$, with an isotopic abundance of < 1 ppt is ignored in all calculations.
   -    The relative amounts of multiply-substituted minor isotopologues are assumed to be in statistical relative abundance, i.e. there is no isotope clumping. Clumping refers to the case where the enrichment (or depletion) of two or more

isotopes in a multiply-substituted isotopologue are correlated rather than each following their statistical amounts independently. Clumping effects are normally much less than 1‰, and according to Table 5 therefore insignificant.

- $^{17}r$ and $\delta^{17}O$ are calculated from $^{18}r$ and $\delta^{18}O$ (Eq. (6)) assuming mass dependent fractionation. Thermodynamic and kinetic fractionation processes are mass-dependent and account for most fractionation mechanisms in nature. Mass-independent fractionation typically occurs in quantum processes such as photolysis and can cause small deviations from mass dependence. These deviations are also typically < 1‰ (e.g. Miller et al., 2002) and thus also negligible for the purposes of this work.

## 7     Other molecules

Similar considerations apply to other molecular species, see Table 6. For $CH_4$, $^{13}CH_4$ measurements are commonly made using laser analysers such as that of Picarro (Rella et al., 2015), and isotopic reference gases are available. An analysis similar to that in section 6 and Table 5 shows that for 2000 ppb $CH_4$ in air, an error of 10‰ in the assumed value of $\delta^{13}C$ leads to an error of 0.2 ppb in the calculated total $CH_4$ mole fraction, and for a -35‰ error the total $CH_4$ error is 0.7ppb. A 100‰ error in $\delta^2H$ leads to an error in total $CH_4$ of only 0.1 ppb.

For $N_2O$ there is the additional complication of the isotopomers $^{15}N^{14}N^{16}O$ and $^{14}N^{15}N^{16}O$ for which standard reference gases are not available, and for which measurement technologies are currently less advanced. The general magnitude of potential errors will be similar to those of $CO_2$. For CO, reference gases are available, but current optical techniques are not able to resolve isotopic variations with sufficient accuracy at the typical low total mole fractions in air.

## 8     Calibration of commercially-available analysers

Several commercial manufacturers offer isotopologue-specific optical analysers based on laser (Campbell, Picarro, Los Gatos Research, Aerodyne, Thermo Fisher Delta Ray) or FTIR (Ecotech Spectronus) spectroscopy that analyse sampled air for one or more specific isotopologues. These instruments report results in a variety of ways, as isotopologue mole fractions and/or as total mole fractions and isotopic delta values, both calibrated and uncalibrated. In most cases the scheme by which total mole fractions and delta values are calculated from the raw measured data is not fully described, although some details are available in user manuals and published works. In most cases some level of approximation is used in accounting for the full molecular isotopic composition when converting between isotopologue amounts and total amounts and delta values. As shown in section 6, these approximations are fortunately in most cases acceptably small, but it is nevertheless recommended that they be assessed and documented if the full computation scheme is not used or measurement and calibration data for all isotopologues are not available.

GAW Greenhouse Gas Measurement Techniques reports since 2011 (WMO-GAW, 2012) recommend that the computational scheme for isotopic quantities derived from all commercial and non-commercial analysers be published and fully transparent to the user to avoid the potential for biases and inaccuracies stemming from different calibration and calculation schemes. Potential errors and calibration biases due to inconsistent isotopic calculations and the empirical determination of concentration dependences can be avoided if only the raw output isotopologue amounts from the analyser(s) are used and calibrated and isotopic quantities are calculated a posteriori following consistent calculation schemes such as those described here and in Flores et al. (2017) and Tans et al. (2017).

## 9    Summary, discussion and conclusions

Optical trace gas analysers based on laser or FTIR spectroscopy measure the concentrations or mole fractions of individual isotopologues of a trace gas rather than the total amount of all possible isotopologues of the target gas. This leads to potential calibration inaccuracies in relating the individual isotopologue measurements made by the analyser to the more usual quantities
of total amount and isotopic ratios or delta values. This paper reviews previous studies addressing isotopic calibration of optical analysers and presents a practical guide to the calculations required to completely and rigorously account for the isotopic composition of a trace gas when determining its total concentration with an isotopologue-specific optical analyser. Although most previous work has made some level of approximation in accounting for the full isotopic composition, this paper shows that such approximations are not required and save little effort - the complete calculations are relatively straightforward. The
approach described here is consistent with those of Flores et al. (2017) and Tans et al. (2017); for $CO_2$ for example, the measurement of either three isotopologues ($^{12}C^{16}O_2$, $^{13}C^{16}O_2$, , $^{12}C^{16}O^{18}O$), or total $CO_2$ and two delta values ($\delta^{13}C$, $\delta^{18}O$) is necessary and sufficient to specify the complete isotopic composition with sufficient accuracy to meet GAW compatibility goals. Calculations to interconvert between these equivalent specifications of composition accurately are described.

Potential errors which may arise when making sometimes-unavoidable approximations in the calculations are assessed and in most cases fortunately found to be small, and often negligible. However significant errors can arise when the isotopic composition of an air sample is very different from that used to calibrated the analyser. Two common cases where this may occur in practice are in the production of synthetic reference standards using highly depleted $^{13}C$ in $CO_2$, and in environmental studies such as soil chambers where high levels of $^{13}C$-depleted $CO_2$ are analysed with an analyser calibrated around clean
atmospheric $^{13}C$ levels.

Provided the appropriate calibration standards are available, this paper recommends that the calibration of optical analysers be carried out via direct measurement of the amounts of individual isotopologues, from which the total trace gas amount and isotopic composition can then be calculated completely and accurately. It recommends against ratio or delta-based calibration
because this approach leads inevitably to concentration dependences in the calibration that must be characterised. Direct isotopologue calibration avoids concentration dependence and requires only reference standards spanning the range of concentrations to be measured and of known isotopic composition. There is no requirement for the reference gases to span the range of expected delta values, they can all be produced from the same source of trace gas and all have the same isotopic composition.

Optical FTIR and laser methods do not currently meet GAW requirements for repeatability of $\delta^{13}C$ in $CO_2$ in clean background air measurements (0.01‰). Their precision is currently limited by the inherent signal:noise ratio of the optical measurement, not by the calibration methodology. The precision currently available from optical measurements is nevertheless very useful for continuous analysis of air in non-baseline scenarios such as urban air or agricultural flux measurements.

## 10    Acknowledgements

I would like to thank the GAW-GGMT community for many discussions on this topic, and especially Edgar Flores, Joelle Viallon, Camille Yver-Kwok and Grant Forster who provided comments on the manuscript and checked the calculations.

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

**Tables**

Table 1. Standard isotope ratios for relevant reference scales used in atmospheric trace gas analysis. [1](Werner and Brand, 2001),
[2](Brand et al., 2010), [3](Bievre et al., 1984), [4] https://www.cfa.harvard.edu/hitran/molecules.html

| Element | Ratio | VPDB[1] | VPDB-CO$_2$ [2] | Hitran[3,4] | VSMOW[1] | air N$_2$[1] |
|---|---|---|---|---|---|---|
| C | $^{13}r = {}^{13}C/{}^{12}C$ | 0.0111802 | 0.0111802 | 0.0112374 | | |
| O | $^{18}r = {}^{18}O/{}^{16}O$ | 0.0020672 | 0.00208835 | 0.0020052 | 0.00200518 | |
| O | $^{17}r = {}^{17}O/{}^{16}O$ | 0.000386 | 0.0003931 | 0.0003729 | | |
| N | $^{15}r = {}^{15}N/{}^{14}N$ | | | 0.00367 | | 0.0036782 |
| H | $^{2}r = {}^{2}H/{}^{1}H$ | | | 0.000156 | 0.00015575 | |

Table 2. Isotopologue fractional abundances and isotopic sums for the VPDB-CO₂ and Hitran scales and conversion factors.
Abundances are taken from [1]*Rothman et al (2005) a*nd [2]https://www.cfa.harvard.edu/hitran/molecules.html for Hitran and [3] *Brand et al.* (2010) for VPDB-CO2. The Brand et al. values supersede earlier values given by *Allison et al. (1995).*

| Isotopologue | Notation | Abundance[1,2] $x_{HITRAN}$ | Abundance[3] $x_{VPDB-CO2}$ | Rescaling factor (Hitran / VPDB-CO2) |
|---|---|---|---|---|
| $^{16}O^{12}C^{16}O$ | 626 | 0.98420 | 0.984054 | 1.000150 |
| $^{16}O^{13}C^{16}O$ | 636 | 0.01106 | 0.0110019 | 1.005280 |
| $^{16}O^{12}C^{18}O$ | 628 | 0.0039471 | 0.00411009 | 0.960319 |
| $^{16}O^{12}C^{17}O$ | 627 | 0.000734 | 0.00077366 | 0.948734 |
| $R_{sum}$ | - | 1.016205 | 1.016053 | 0.9998505 |

**Table 3. Worked data for calibration of an FTIR analyser using four reference standards (a) using actual mole fractions of all isotopologues, and (b) using normalised mole fractions on the VPDB-CO₂ scale. $^{17}r$ and $\delta^{17}O$ values were not directly determined and are not included in the table – they are derived from $^{18}r$ and $\delta^{18}O$ following equation (6).**

| (a) Tank | $y_{CO_2}$ ppm | $\delta^{13}C$ ‰ | $\delta^{18}O$ ‰ | $^{13}r$ | $^{18}r$ | $R_{sum}$ | $y_{626}$ ppm | $y_{636}$ ppm | $y_{628}$ ppm |
|---|---|---|---|---|---|---|---|---|---|
| | **Calibration tank data** | | | | | | **Reference mole fractions** | | |
| CB11138 | 396.74 | -8.38 | 0.30 | 0.011087 | 0.002089 | 1.016112 | 390.45 | 4.3287 | 1.6313 |
| CB11483 | 452.06 | -8.19 | -2.11 | 0.011089 | 0.002084 | 1.016103 | 444.90 | 4.9333 | 1.8543 |
| CA06845 | 416.06 | -10.69 | -2.71 | 0.011061 | 0.002083 | 1.016072 | 409.48 | 4.5291 | 1.7056 |
| CB09950 | 392.91 | -8.38 | -0.20 | 0.011087 | 0.002088 | 1.016110 | 386.68 | 4.2870 | 1.6147 |
| | | | | | | | **Measured mole fractions** | | |
| CB11138 | | | | | | | 426.50 | 4.9011 | 1.8942 |
| CB11483 | | | | | | | 486.46 | 5.5937 | 2.1768 |
| CA06845 | | | | | | | 447.49 | 5.1310 | 1.9891 |
| CB09950 | | | | | | | 422.33 | 4.8533 | 1.8731 |
| | | | | | | | **Calibration coefficients** | | |
| | | | | | | **Slope a** | 1.10146 | 1.14563 | 1.26747 |
| | | | | | | **Intercept b** | -3.56 | -0.0579 | -0.1733 |

| (b) Tank | $y_{CO_2}$ ppm | $\delta^{13}C$ ‰ | $\delta^{18}O$ ‰ | $^{13}r$ | $^{18}r$ | $X_{sum}$ | $y'_{626}$ ppm | $y'_{636}$ ppm | $y'_{628}$ ppm |
|---|---|---|---|---|---|---|---|---|---|
| | **Calibration tank data** | | | | | | **Reference normalised mole fractions** | | |
| CB11138 | 396.74 | -8.38 | 0.30 | 0.011087 | 0.002089 | 0.999909 | 396.78 | 393.45 | 396.90 |
| CB11483 | 452.06 | -8.19 | -2.11 | 0.011089 | 0.002084 | 0.999900 | 452.11 | 448.40 | 451.15 |
| CA06845 | 416.06 | -10.69 | -2.71 | 0.011061 | 0.002083 | 0.999869 | 416.11 | 411.67 | 414.99 |
| CB09950 | 392.91 | -8.38 | -0.20 | 0.011087 | 0.002088 | 0.999906 | 392.95 | 389.66 | 392.87 |
| | | | | | | | **Measured normalised mole fractions** | | |
| CB11138 | | | | | | | 433.414 | 445.48 | 460.87 |
| CB11483 | | | | | | | 494.342 | 508.43 | 529.62 |
| CA06845 | | | | | | | 454.742 | 466.38 | 483.96 |
| CB09950 | | | | | | | 429.172 | 441.13 | 455.74 |
| | | | | | | | **Calibration coefficients** | | |
| | | | | | | **Slope a** | 1.10146 | 1.14563 | 1.26748 |
| | | | | | | **Intercept b** | -3.62 | -5.27 | -42.16 |

**Table 4. Worked calibration of sample data in Figure 3 at four times with varying CO₂ mole fractions. Columns 2-4 contain the raw measured isotopologue mole fractions corrected to dry air, columns 5-7 contain the calibrated dry air mole fractions after applying the coefficients from Table 3, columns 8-10 are the isotopic ratios and $R_{sum}$ for each sample, and columns 11-13 contain the final calibrated total CO₂, $\delta^{13}C$ and $\delta^{18}O$.**

| Time 23/24 Jan | $y_{626,meas}$ ppm | $y_{636,meas}$ ppm | $y_{628,meas}$ ppm | $y_{626,cal}$ ppm | $y_{636,cal}$ ppm | $y_{628,cal}$ ppm | $^{13}r$ | $^{18}r$ | $R_{sum}$ | $y_{CO2}$ ppm | $\delta^{13}C$ ‰ | $\delta^{18}O$ ‰ |
|---|---|---|---|---|---|---|---|---|---|---|---|---|
| 18:00 | 433.79 | 4.9845 | 1.9325 | 397.07 | 4.4015 | 1.6614 | 0.011085 | 0.002092 | 1.016117 | 403.47 | -8.53 | 1.76 |
| 00:00 | 492.97 | 5.6550 | 2.2211 | 450.80 | 4.9867 | 1.8891 | 0.011062 | 0.002095 | 1.016102 | 458.05 | -10.56 | 3.31 |
| 06:00 | 541.01 | 6.2000 | 2.4531 | 494.41 | 5.4624 | 2.0722 | 0.011048 | 0.002096 | 1.016088 | 502.37 | -11.80 | 3.46 |
| 12:00 | 433.37 | 4.9800 | 1.9309 | 396.69 | 4.3975 | 1.6601 | 0.011086 | 0.002092 | 1.016119 | 403.08 | -8.47 | 1.97 |

**Table 5. Actual isotopologue amounts and $R_{sum}$ values in 400 ppm total CO₂ for various isotopic compositions. The last column lists errors in calculating total CO₂ if different isotopic composition between reference (calibration) and sample measurements are not accounted for. See text for details of the various cases.**

| Case | $y_{CO2}$ ppm | $\delta^{13}C$ ‰ | $\delta^{18}O$ ‰ | $\delta^{17}O$ ‰ | $R_{sum}$ | $y_{626}$ ppm | $y_{636}$ ppm | $y_{628}$ ppm | $y_{CO2}$ error ppm |
|---|---|---|---|---|---|---|---|---|---|
| 1 | 400 | 0 | 0 | 0 | 1.01620 | 393.62 | 4.4077 | 1.6440 | 0.000 |
| 2 | 400 | −8 | 0 | 0 | 1.01611 | 393.66 | 4.3660 | 1.6442 | 0.035 |
| 3 | 400 | −35 | 0 | 0 | 1.01581 | 393.77 | 4.2484 | 1.6447 | 0.155 |
| 4 | 400 | 0 | 2 | 0 | 1.01621 | 393.62 | 4.4007 | 1.6473 | −0.003 |
| 5 | 400 | 0 | 0 | 2 | 1.01621 | 393.62 | 4.4007 | 1.6440 | −0.001 |
| 6 | 400 | 5.13 | −39.82 | −51.4 | 1.01605 | 393.68 | 4.4240 | 1.5788 | 0.060 |
| 7 | 400 | 0 | 0 | 0 | 1.01614 | 393.65 | 4.4010 | 1.6441 | 0.024 |

**Table 6. Details of isotopologues of common atmospheric species.**

| Species | Stable isotopes | No. isotopocules Total (distinct) | No. independent quantities to specify isotopic composition | $R_{sum}$ |
|---|---|---|---|---|
| $CO_2$ | $^{12}C$, $^{13}C$ $^{16}O$, $^{17}O$, $^{18}O$ | 18 (12) | 4 | $(1+^{13}r).(1+^{17}r+^{18}r)^2$ |
| $CH_4$ | $^{12}C$, $^{13}C$ $^{1}H$, $^{2}H$ | 32 (10) | 3 | $(1+^{13}r).(1+^{2}r)^4$ |
| $N_2O$ | $^{14}N$, $^{15}N$ $^{16}O$, $^{17}O$, $^{18}O$ | 12 (12) | 4 | $(1+^{15}r)^2.(1+^{17}r+^{18}r)$ |
| $CO$ | $^{12}C$, $^{13}C$ $^{16}O$, $^{17}O$, $^{18}O$ | 6 (6) | 4 | $(1+^{13}r).(1+^{17}r+^{18}r)$ |
| $H_2O$ | $^{1}H$, $^{2}H$ $^{16}O$, $^{17}O$, $^{18}O$ | 12 (9) | 4 | $(1+^{2}r)^2.(1+^{17}r+^{18}r)$ |

**Figures and Figure captions**

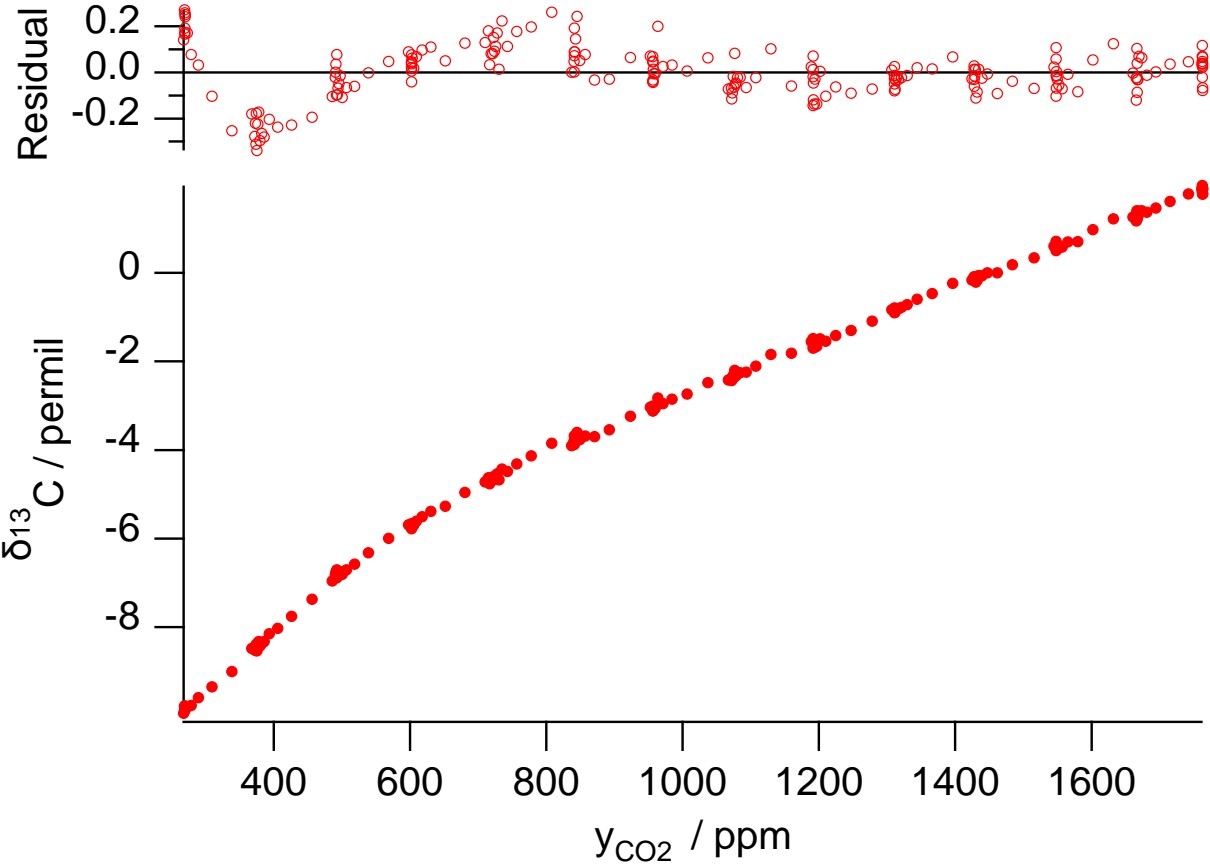

**Figure 1. Example of $\delta^{13}C$ dependence on $CO_2$ mole fraction for a Spectronus FTIR analyser. The measured data are fitted with a function of form of Eq. (15) with fitted parameters $\beta$ = - 1227 ‰ ppm and $\gamma$ = 0.0054 ‰ ppm$^{-1}$.**

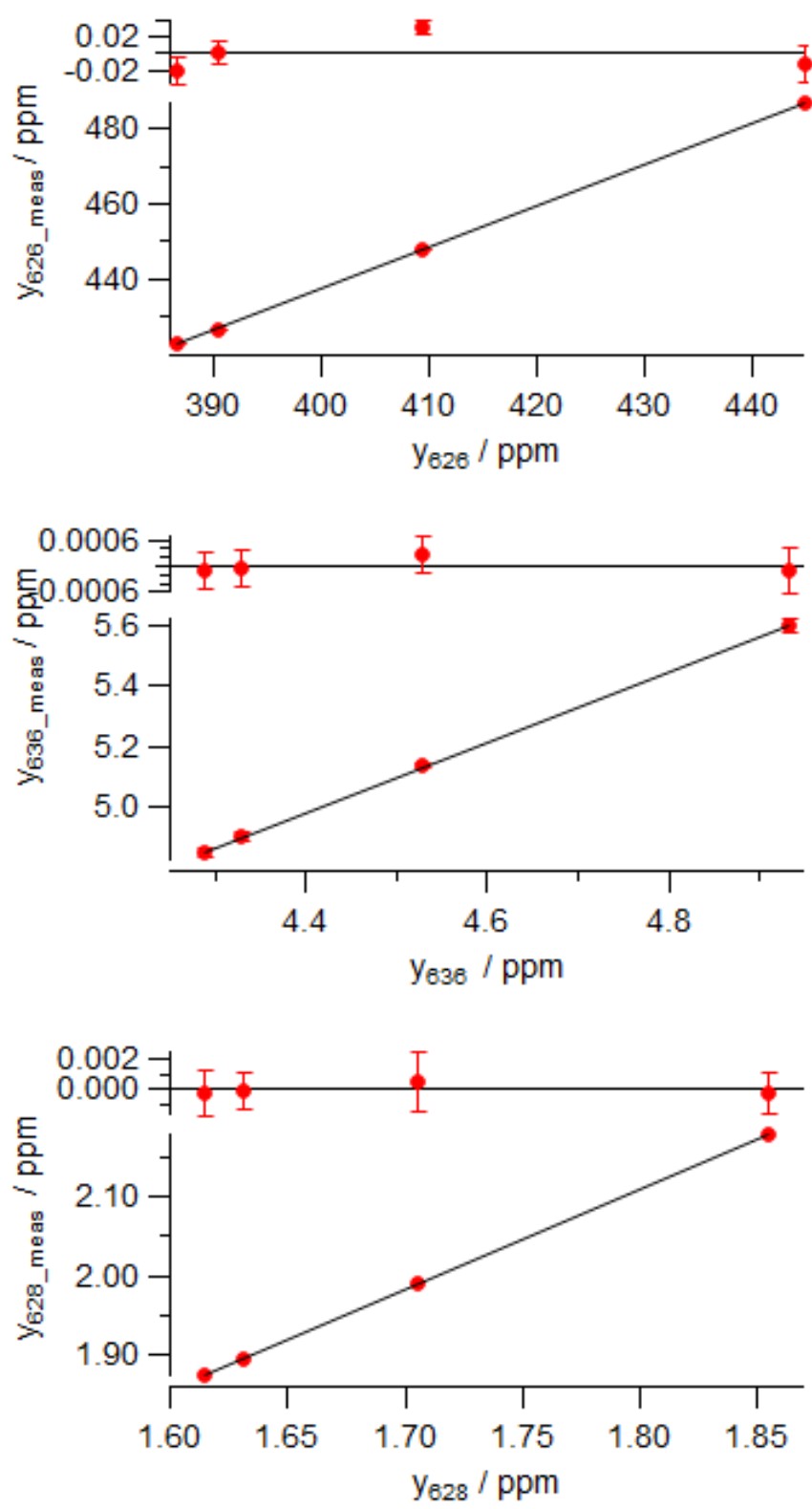

**Figure 2. Calibration plots for three CO₂ isotopologues.**

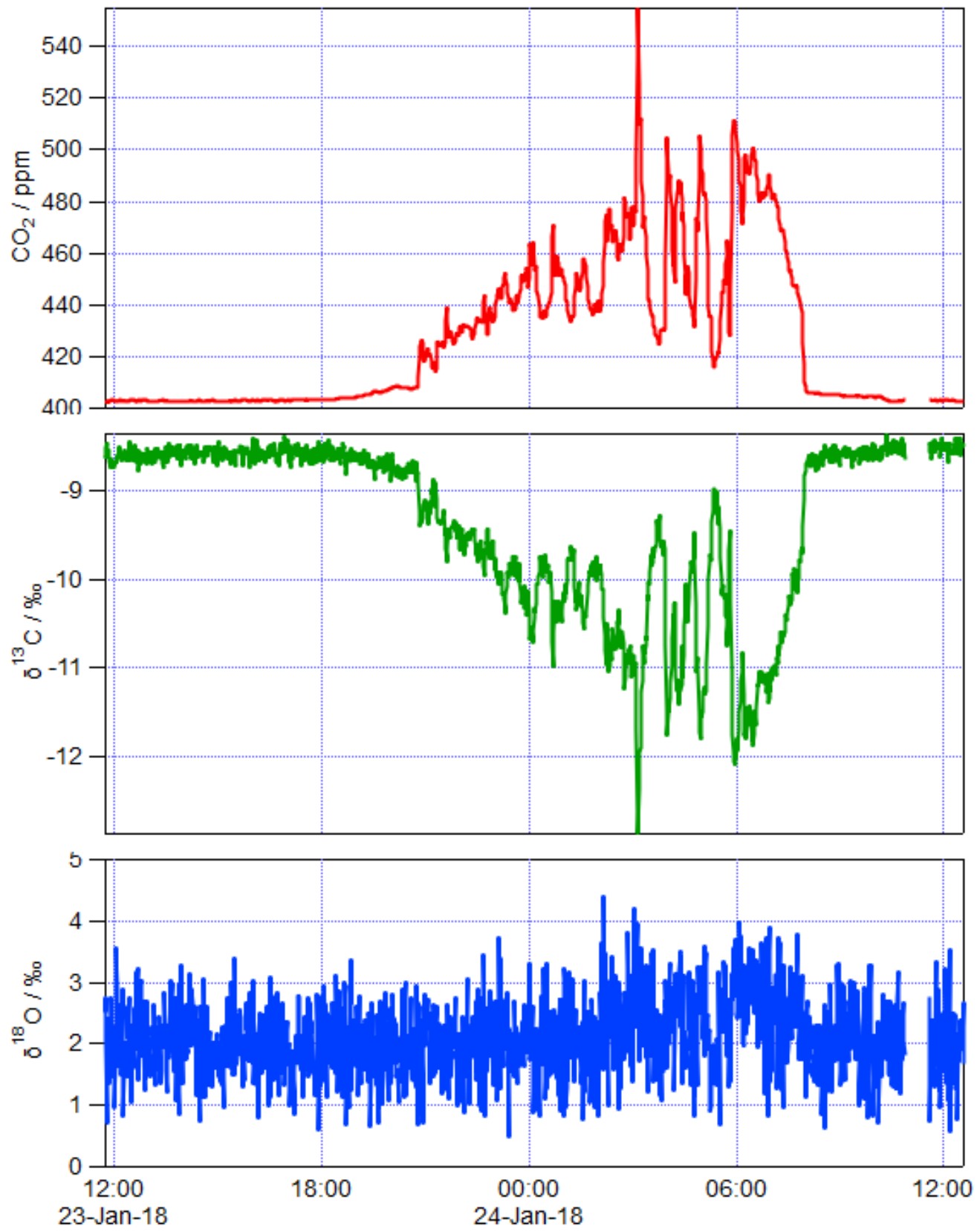

**Figure 3. Calibrated total CO₂, δ¹³C and δ¹⁸O of sampled air on 23-24 Jan 2018 at a rural site in SE Australia. Air was sampled continuously, the displayed data are 1 minute averages.**

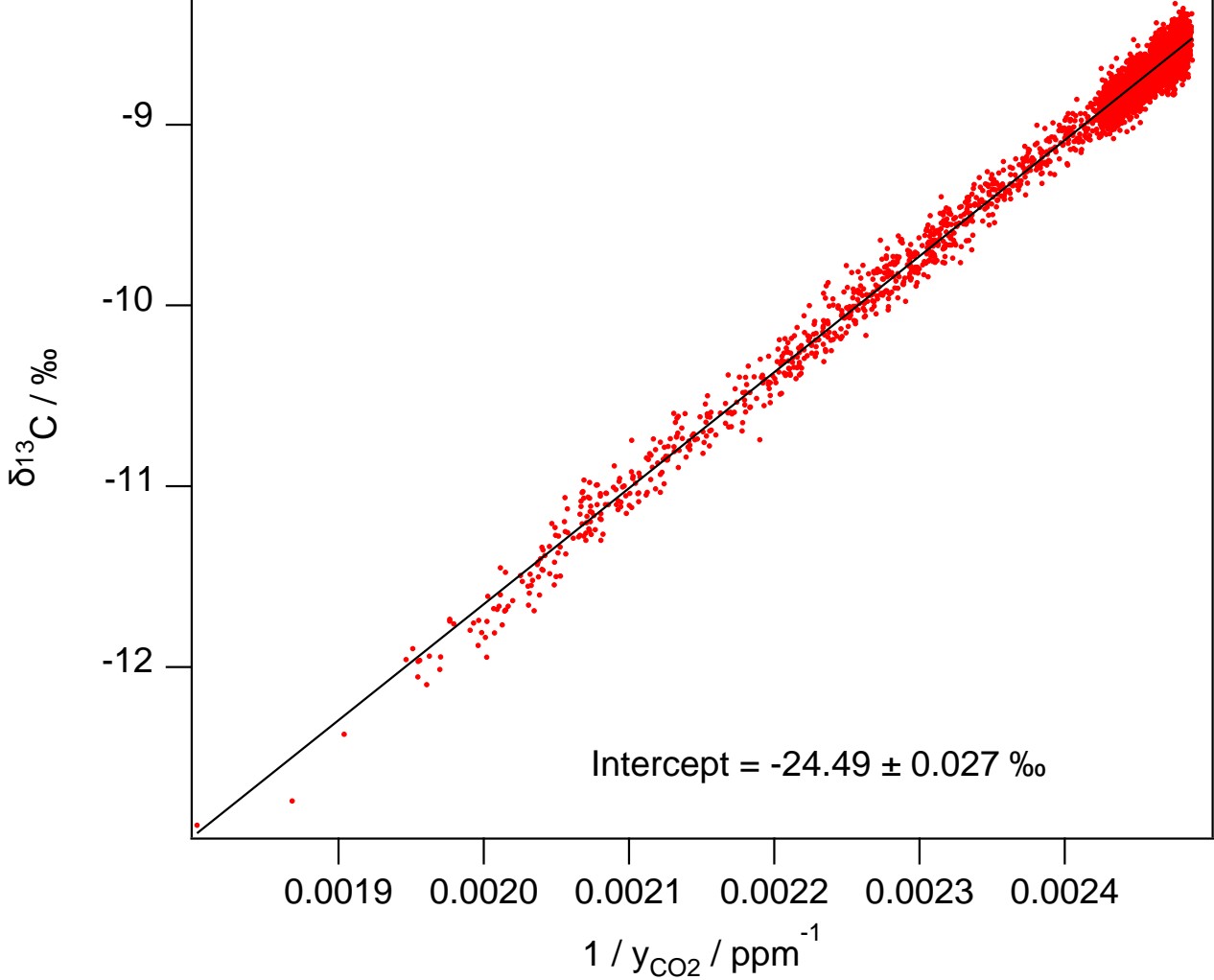

**Figure 4. Keeling plot of data shown in Figure 2.**

