# Peer review of "Calibration of isotopologue-specific optical trace gas analysers: A practical guide"

_Atmospheric Measurement Techniques, 2018_

## Short Comment (SC1) · 5 Jul 2018

It is nice to see this practical guide to calibration of optical isotopologue spectrometers, which gives much needed careful consideration to the topic. I am sure it will be helpful to the research community. In order to fully contextualize this work, I would just recommend that the author consider and cite Wehr et al., "Long-term eddy covariance measurements of the isotopic composition...", Agricultural and Forest Meteorology 181 (2013) 69–84. Section 3.5 and Appendix A of Wehr et al. describe and advocate the same isotopologue-specific calibration approach as the present manuscript, including full accounting of all non-negligible isotopologues. In particular, the present manuscript's Section 2 is effectively identical to Wehr et al.'s Appendix A.

[Figure]

Wehr et al. also raises the issue of cross-talk between the isotopologue spectral lines (i.e. when the concentration that the instrument reports for one isotopologue depends on the concentration of another isotopologue), which can occur due to partial overlap of the lines and inadequate spectral fitting. Cross-talk was not a problem for our particular instrument but could induce substantial error in general when using the isotopologue calibration approach. It might be a good idea to raise this point in the present guide, as it is something users of these instruments ought to check for.

Regards,

Rick Wehr

―――――――――――――――――

---

## Author Comment (AC1) · 6 Jul 2018

I would like to thank Rick Wehr for this short comment and for bringing this paper to my attention – it did not come up in any of my searches, nor did I find it referenced by other papers in the area. I agree that Appendix A is indeed fully consistent in its treatment of the isotopologue calculations and should be duly acknowledged. I will cite the paper in the revised manuscript and include it in the description of prior work.
* * *

---

## Referee Comment (RC1) · Anonymous Referee #1 · 13 Aug 2018

Review of:

**Calibration of isotopologue-specific optical trace gas**
**analysers: A practical guide**

by David W. T. Griffith

**General comments:**

The manuscript is well written and very helpful for someone anyone working with isotopologue-
specific optical trace gas analysers. Calibrating such instruments (or understanding the calibration)
is a crucial element in this field and it fits well that this guide is published to bring everybody on the
same page in this growing community. An important point is flawless calculations in this guide (I
cannot find any) which was taken serious by the author by discussing/checking the calculations
with several other experts in the field – as stated in the acknowledgement. In summary, this is a
useful guide and is recommended to be published after a revision.

I have one major comment that regards the comparison of the two different ways to calibrate optical
instruments and the suggestions made in this regards.

The few minor comments I have are mainly clarifications/deeper explanations about the underlying
idea/calculations. The manuscript has an appreciated educational character and seems also to aim on
students, so it might be worth to elaborate the mathematical idea a bit more at some points.

**Major comment:**

The author recommends using the "direct calibration by isotopologues amounts" due to its apparent
advantages (no pressure dependence, the need of only one reference gas in different "dilutions") as
opposed to the calibration via different reference gases with different δ-value ("Calibration by delta
values"). I think this recommendation can only be done if the targeted precision in δ-value is not
exceeding the precision of the absolute concentrations of the reference gas and its "dilutions". For
example, the precision of $CO_2$ absolute concentration in reference gases from international
reference labs is around 0.07 ppm[1]. The same type of labs can provide the $\delta^{13}C(CO_2)$ in such gases
using IRMS with a precision in the 0.02‰ range or better, while this δ-value is relative to a material
with a specific isotopic composition (reference material) and therefore independent of the absolute
concentration of $CO_2$ in that gas. The 0.02‰ in $\delta^{13}C(CO_2)$ corresponds to a precision in absolute
concentration of about 0.008 ppm (for a 400 ppm concentration gas), so about one order of
magnitude better than the absolute concentration actually can be known (in absolute, SI-traceable
fashion).

As far I understand the recommended calibration approach, it ultimately depends on the precision of
the absolute $CO_2$ concentration of the used gases. Since the current limit of these values is around
0.07 ppm, the corresponding precision in $\delta^{13}C(CO_2)$ cannot be better than about 0.2‰ (exceptions
exist if the measured value is very close to the used reference gas in the calibration). If the user
dilutes its reference gas in a self-made experiment, it's very likely that the dilution process will
introduce additional uncertainty making the in $\delta^{13}C(CO_2)$ precision (and obviously also the
precision of the absolute $CO_2$ concentration) worse.

This disadvantage of the "direct calibration by isotopologues amounts" has to be addressed in the
manuscript and the suggestion has to be adapted accordingly. A 0.2‰ in $\delta^{13}C(CO_2)$ is one order of
magnitude above the GAW recommendations[2] for which reason the proposed calibration method
seems not applicable for instruments applied in the context of GAW. Some of the references used in
the manuscript as an example of the alternative "Calibration by delta values" are used in the GAW
context with corresponding precision targets. So the context of these references has to be
reconsidered.

**Minor comments:**

-Clumped isotopes are mentioned a few times in the manuscript, but there is no example of it. It
would be nice to have an example for the uneducated reader. In general, one could consider to
extend table 2 with all 18 (or 12 distinct) possible isotopologues as an overview and list its
abundances (as much as known) as an overview.

-Page 4, line 2 and following formulas: you shorten from $n(^{12}C)$ to $^{12}C$. I recommend to not do this
because it complicates the comparison with the previous formulas and the shortening is only used in
the following formulas (as much I can see).

-Page 4, line 11-16: This section is not very clear to me. You only take 4 distinct isotopologues into
consideration, but $R_{sum}$ seems to be the sum of all 18 (according to the current text). Or how do I
have to understand this? $R_{sum}$ seems to be a rather abstract parameter, or does it have any easy-
understandable meaning?

-Page 7, line 35 and following: IRMSs can also be used to measure absolute abundances of the
single isotopologues, but the measurement is in general noisy and very difficult to calibrate. This
"absolute" approach has been used in the beginning of mass spectrometry before it has been
realized that a relative measurement with IRMS is much more precise than an absolute
measurement[3]. This is also the case for isotopologue-specific optical systems. The noise in a line
ratio is much smaller than the noise of the individual lines because there is a lot of
correlated/technical noise on two simultaneously measured lines which cancels out in a ratio (given
that the instrument can measure the targeted lines simultaneously in real-time as possible e.g. with a
direct absorption spectrometer). It's not due to the nature of IRMS that the community has
converged to go for relative δ-scales, but rather due to the more fundamental analytical advantage
of measuring relative values with respect to a reference material as opposed to absolute
measurements.

**References:**

1.    Zhao CL, Tans PP. Estimating uncertainty of the WMO mole fraction scale for carbon
dioxide in air. *J Geophys Res Atmos*. 2006;111(8):1-10. doi:10.1029/2005JD006003

2.    WMO. 18th WMO/IAEA meeting on carbon dioxide, other greenhouse gases and related
measurement techniques (GGMT-2015). *GAW Rep No 229*. 2016;(229):13-17.
https://library.wmo.int/opac/doc_num.php?explnum_id=3074.

3.    Criss RE. *Principles of Stable Isotope Geochemistry*. Oxford University Press; 1999.
doi:10.1016/S0037-0738(97)00056-0

---

## Referee Comment (RC2) · Anonymous Referee #2 · 24 Sep 2018

This short paper is, I think, a welcome and highly useful guide. It should be helpful to both researchers in the field commencing measurements with new isotopologue-specific analysers and also to students. The paper is well written and has a level of detail that is fit-for-purpose - as a practical guide. The worked examples are a nice addition, especially for students.

My one general comment would be that although the approach taken by the author does indeed side-step the concentration dependence arising if one calibrates using a ratio-based approach, IRMS methods arose because of the precision advantage arising when measuring a ratio. It is certainly still the case that the precision with which any of the isotopologue-specific optical instruments can measure the minor isotopologues is limited 0.1 ppm (generously), so this will impact the precision with which one

can determine a delta value from these instruments via this otherwise straightforward approach. As such, there remain advantages in working with the ratio even with instruments that 'naturally' lend themselves to this approach. Some acknowledgement of this ought to be made in the text.

Specific comments:

P 3, line 12: Clumped isotopes are mentioned, but not described, but as I see great value in this manuscript for students, I think it would be helpful to extend the parenthese to something like: (i.e. clumped isotopes, is the term for isotopologues carrying two (or more) of the heavy, rare isotopes.)

p 9, line11: 2018 should read 2017.

p 9. line 25: I think case 7 needs a little more explanation to be clear to readers.

---

## Author Comment (AC2) · 11 Oct 2018

**Author comments, Anonymous referee #1**

I thank referee 1 for perceptive and constructive comments and make the following responses:

**Major comment:**

*The author recommends using the "direct calibration by isotopologues amounts" due to its apparent advantages (no pressure dependence, the need of only one reference gas in different "dilutions") as opposed to the calibration via different reference gases with different $\delta$-value ("Calibration by delta values"). I think this recommendation can only be done if the targeted precision in $\delta$-value is not exceeding the precision of the absolute concentrations of the reference gas and its "dilutions". For example, the precision of $CO_2$ absolute concentration in reference gases from international reference labs is around 0.07 ppm[1]. The same type of labs can provide the $\delta_{13}C(CO_2)$ in such gases using IRMS with a precision in the 0.02‰ range or better, while this $\delta$-value is relative to a material with a specific isotopic composition (reference material) and therefore independent of the absolute concentration of $CO_2$ in that gas. The 0.02‰ in $\delta_{13}C(CO_2)$ corresponds to a precision in absolute concentration of about 0.008 ppm (for a 400 ppm concentration gas), so about one order of magnitude better than the absolute concentration actually can be known (in absolute, SI-traceable fashion).*
*As far I understand the recommended calibration approach, it ultimately depends on the precision of the absolute $CO_2$ concentration of the used gases. Since the current limit of these values is around 0.07 ppm, the corresponding precision in $\delta_{13}C(CO_2)$ cannot be better than about 0.2‰ (exceptions exist if the measured value is very close to the used reference gas in the calibration). If the user dilutes its reference gas in a self-made experiment, it's very likely that the dilution process will introduce additional uncertainty making the in $\delta_{13}C(CO_2)$ precision (and obviously also the precision of the absolute $CO_2$ concentration) worse.*
*This disadvantage of the "direct calibration by isotopologues amounts" has to be addressed in the manuscript and the suggestion has to be adapted accordingly. A 0.2‰ in $\delta_{13}C(CO_2)$ is one order of magnitude above the GAW recommendations for which reason the proposed calibration method seems not applicable for instruments applied in the context of GAW. Some of the references used in the manuscript as an example of the alternative "Calibration by delta values" are used in the GAW context with corresponding precision targets. So the context of these references has to be reconsidered.*

**Response:**

This is a valid comment, also raised by referee 2, but highlights some misunderstandings which clearly should be clarified.

Firstly there is the minor misconception that the reference standards for absolute isotopologue calibration are obtained by dilution of a high-level standard.  This is not usually the case, the reference gases are normally obtained as whole air mixtures individually calibrated by a certified GAW laboratory (such as NOAA, ICOS or Gaslab) with a range of CO2 mole fractions but in a matrix of other gases (N2, O2, Ar, CH4,, CO, N2O) which are typical of atmospheric composition. Thus there is no loss of accuracy due to an in-lab dilution process. If standards are prepared in the reference gas supply lab over a range of mole fractions using the same CO2 source gas the delta values will be the same for all standards over the range of mole fractions – this was the case in Flores et al. Such standards are easier to prepare but suitable only for isotopologue-based calibration.

The referee points out that for reference standards with absolute mole fractions known to 0.07 ppm (0.18‰), the best possible absolute accuracy for $\delta^{13}C$ would be ~0.2‰. However this is only true if the individual isotopologue mole fractions are each individually known to 0.18‰ with independent errors(the uncertainty in $\delta^{13}C$ would actually be sqrt(2) greater, or 0.25‰ because there are two measurements in the calculation of delta). In practice the calibration lab provides a total CO2 amount fraction (accuracy say 0.18‰) and $\delta^{13}C$ value (accuracy say 0.02‰). Following the calculations in this paper, the 636/626 ratio is as accurately known as the reference $\delta^{13}C$ value provided (not the total $CO_2$ accuracy), and the individual isotopologue values as accurately as the total CO2 accuracy. The accuracy of the isotope ratio provided by the calibration lab is preserved through the isotopic calculations and the calibration process.

Now consider the optical measurement, whether FTIR or laser-based. Using the 1-min Allan deviation figures from Griffith et al (2012) as an example, we take a repeatability for CO2 isotopologues of 0.02 ppm (0.05‰) and for $\delta^{13}C$ we take 0.07‰ (which is approximately 0.05*sqrt(2) ‰). For laser instruments the figures are similar. If errors in retrieving the individual isotopologue amounts from the spectra were to some degree correlated, the $\delta^{13}C$ repeatability would be better. (This may be more the case for laser instruments analysing single lines than for FTIR analysing whole spectral bands, as pointed out in a specific comment by this referee). Thus the *repeatability* of the spectral $\delta^{13}C$ measurement in this example is of the order of 0.07‰, but its *accuracy* is determined by the calibration against reference standards, which is governed by the reference $\delta^{13}C$ values and may be better (ie the measurement of delta may be more accurate than it is precise). The ultimately reported accuracy of the sample delta value will be limited by the largest source of error, the isotopologue measurement repeatabilities, and thus of the order of 0.07‰. It is not determined by the absolute accuracy of the total CO2 amounts in the reference gases. This accuracy is still larger than the GAW-GGMT requirement, but is the current state of the art for optical measurements.

Finally, it is important to recognise that both the absolute and the ratio calibration methods take a ratio of 636/626 to determine $^{13}r$ and $\delta^{13}C$ and thus both take advantage of the accuracy of the isotope ratio calibration of the reference gases. There is no equivalent in optical measurements to IRMS, where the instrumentation primarily measures a ratio of ion counts, but not individual amounts. In this sense both methods are equally precise.The difference between the absolute and ratio calibration of optical measurements lies in whether the calibrated or uncalibrated 636/626 ratio is used. In the former case (ratio of calibrated 636 and 626) the calibrated $\delta^{13}C$ is calculated directly from the calibrated isotopologue amounts, but in the latter (ratio calibration) a range of $\delta^{13}C$ in reference gases is required, and there is an unavoidable concentration dependence (Eq. 15) which must be characterised. The form of eq 15 suggests a 3-parameter fit to characterise this concentration dependence, but the coefficients $\beta$ and $\gamma$ are not actually constants so the parameterisation is only approximate, and the values of $\beta$ and $\gamma$ are related to the instrument calibration and should be in principle re-determined with every calibration. In my FTIR experience, the residuals in fitting data to Eq 15 may result in residuals as large as 0.5‰. Other instruments may take a different approach to determining this concentration dependence, but it is likely to add error to the determined delta values, and can never reduce the error.

To address these comments I have extended the discussion of ratio calibration in section 4.2 to include a figure illustrating the concentration dependence through application of Eq. 15 and the potential errors using real FTIR data, and I have added a new section 4.3 which is a condensed form of the response given above.

Added to 4.2

describe the concentration dependence. The linear term becomes relatively more important than the inverse term at high $CO_2$ mole fractions where the inverse $CO_2$ term becomes small and any quadratic contribution to the calibration equation leading to the linear term becomes large.

Figure 1 illustrates this concentration dependence with a typical $\delta^{13}C$ vs $CO_2$ dependence for an FTIR analyser similar to that used in the example of section **Error! Reference source not found.** below. The dependence was determined by continuous flow measurements of a single $CO_2$-spiked air tank while the $CO_2$ content was gradually reduced by passing a fraction of the flow through Ascarite. The measured $\delta^{13}C$ vs $CO_2$ data are fitted to Eq **Error! Reference source not found.** with fitted parameters $\beta$ = - 1227 ‰ ppm and $\gamma$ = 0.0054 ‰ ppm$^{-1}$, corresponding to $CO_2$ - dependent corrections of up to 5‰ over the $CO_2$ range 400-1000 ppm. The residuals of the fit illustrate potential errors from the modelled behaviour of up to ± 0.3‰. Uncertainties in calibrating the $CO_2$ concentration dependence can lead to significant errors in Keeling-type analyses over a wide range of total $CO_2$ amounts even if the isotopologue calibration non-linearity is very small (Pang et al., 2016; Wen et al., 2013).

[Figure]

**Figure 1. Example of $\delta^{13}C$ dependence on $CO_2$ mole fraction for a Spectronus FTIR analyser. The measured data are fitted with a function of form of Eq.** Error! Reference source not found. **with fitted parameters $\beta$ = - 1227 ‰ ppm and $\gamma$ = 0.0054 ‰ ppm$^{-1}$.**

Added 4.3

**Comments on the accuracy of optical isotopologue and ratio calibration**

As an example assume a calibration laboratory provides calibrated reference gases with an absolute accuracy of 0.05 ppm for total $CO_2$ amount (0.12‰ in 400 ppm $CO_2$) and 0.02‰ for $\delta^{13}C$ measured by IRMS. The isotope ratio is thus more accurately determined than the total amount fraction for the reference gases. Now take as a practical measurement repeatability for optical analysers 0.02 ppm

(0.05‰) for total $CO_2$ amount and 0.07‰ for $\delta^{13}C$ (e.g. Griffith et al. (2012), laser instruments similar). The absolute accuracy for the calibrated optical measurement of total $CO_2$ is limited by the reference gas amount fraction, but the more accurately known reference $^{13}r$ or 626/636 ratio is carried through the calibration calculations and this accuracy is preserved when measured isotopologue amounts are ratioed. The accuracy of measured $^{13}r$ or $\delta^{13}C$ is thus limited by the optical measurement (0.07‰) which is less precise than the IRMS-provided reference accuracy (0.02‰). This does not currently meet GAW requirements for clean background air measurements (WMO-GAW, 2016), but is nevertheless useful for continuous analysis of air in non-baseline scenarios such as urban air or agricultural flux measurements.

This reasoning applies to both isotopologue and ratio calibration schemes, which both benefit from the higher accuracy in the isotopologue ratios than in absolute isotopologue amounts. The principle differences between the isotopologue and ratio calibration schemes are twofold:

- The isotopologue scheme does not require calibration gases spanning a range of delta values, it is sufficient to span the range of total amount fractions of interest. This simplifies the preparation of reference gases for calibration laboratories.
- The ratio scheme has an unavoidable $CO_2$ concentration dependence which must be characterised and leads potentially to a loss of accuracy, as shown in section **Error! Reference source not found.**. This complicates the calibration procedure for optical analysers.

Errors are discussed further in section 6.

**Specific comments**

*Clumped isotopes are mentioned a few times in the manuscript, but there is no example of it. It would be nice to have an example for the uneducated reader. In general, one could consider to extend table 2 with all 18 (or 12 distinct) possible isotopologues as an overview and list its abundances (as much as known) as an overview.*

There is a confusion (from both referees) here between multiply-substituted isotopologues (isotopologues with more than one minor isotope) and clumped isotopes (referring to multiply-substituted isotopologues where the relative amounts of the isotopes are not statistical, eg where two isotopes are both enriched together above their bulk statistical abundances). To introduce clumped isotopes more clearly, as requested, I have amended the text at first usage in section 2:

Using $CO_2$ as an example, considering the stable isotopes $^{12}C$, $^{13}C$, $^{16}O$, $^{17}O$ and $^{18}O$, there are eighteen possible isotopologues (2 x 3 x 3 isotopic possibilities). $^{14}C$ is a negligible proportion of total carbon for these purposes and is neglected. Only twelve of these eighteen possibilities are distinct due to symmetry. Assuming the substitution of each isotope at each position in the molecule follows its bulk statistical abundance (i.e. no clumping, see section **Error! Reference source not found.**), only four independent quantities are required to fully define the total amount and full isotopic composition of $CO_2$. These quantities may equivalently be the total $CO_2$ amount and three isotopic ratios $^{13}r$, $^{17}r$ and $^{18}r$ (or delta values $\delta^{13}C$, $\delta^{17}O$ and $\delta^{18}O$), or the amounts of four individual isotopologues with each isotope substituted, most conveniently $^{16}O^{12}C^{16}O$, $^{16}O^{13}C^{16}O$, $^{16}O^{12}C^{17}O$ and $^{16}O^{12}C^{18}O$. Once these are known, the abundances of all multiply-substituted isotopologues can be calculated.

I have also expanded and clarified this description of "clumping" in section 6:

The relative amounts of multiply-substituted minor isotopologues are assumed to be in statistical relative abundance, i.e. there is no isotope clumping. Clumping refers to the case where the enrichment (or depletion) of two or more isotopes in a multiply-substituted isotopologue are correlated rather than each following their statistical amounts independently. Clumping effects are normally much less than 1‰, and according to **Error! Reference source not found.** therefore insignificant.

*-Page 4, line 2 and following formulas: you shorten from n(₁₂C) to ₁₂C. I recommend to not do this because it complicates the comparison with the previous formulas and the shortening is only used in the following formulas (as much I can see).*

Done:

$$^{12}x = \frac{n(^{12}C)}{n(^{12}C)+n(^{13}C)} = \frac{1}{(1+{}^{13}r)}$$

$$^{13}x = \frac{n(^{13}C)}{n(^{12}C)+n(^{13}C)} = \frac{{}^{13}r}{(1+{}^{13}r)}$$

$$^{16}x = \frac{n(^{16}O)}{n(^{16}O)+n(^{17}O)+n(^{18}O)} = \frac{1}{(1+{}^{17}r+{}^{18}r)}$$

$$^{17}x = \frac{n(^{17}O)}{n(^{16}O)+n(^{17}O)+n(^{18}O)} = \frac{{}^{17}r}{(1+{}^{17}r+{}^{18}r)}$$

$$^{18}x = \frac{n(^{18}O)}{n(^{16}O)+n(^{17}O)+n(^{18}O)} = \frac{{}^{18}r}{(1+{}^{17}r+{}^{18}r)}$$

*-Page 4, line 11-16: This section is not very clear to me. You only take 4 distinct isotopologues into consideration, but $R_{sum}$ seems to be the sum of all 18 (according to the current text). Or how do I have to understand this? $R_{sum}$ seems to be a rather abstract parameter, or does it have any easy- understandable meaning?*

$R_{sum}$ is analogous to a partition sum over all possible energy levels of a molecule and provides a convenient normalising factor when calculating relative isotopologue abundances (cf relative populations in the case of energy levels). In practice it is most easily thought of as the ratio of the total amount of $CO_2$ in a sample to the that of the major isotopologue. For example if Rsum=1.01, 1/Rsum = 0.99 and 99% of the sample is 626. I have expanded the wording in this paragraph as follows to try and make this clearer:

The labels 626, 636, 628, 627 are the common isotopic shorthand used in spectroscopy and the Hitran database. The sum of all isotopologue abundances $x$ over all 18 isotopologues is equal to unity. $R_{sum}$ is a sum of 18 products of isotope ratios, one corresponding to each of the 18 possible isotopologues of $CO_2$. $R_{sum}$ conveniently accounts for all possible isotopologues in calculations of abundances, providing a normalising factor somewhat analogous to a partition sum over all energy levels of a molecule. From Eq. **Error! Reference source not found.**, $x_{626} = 1/R_{sum}$ i.e. $1/R_{sum}$ is the fractional abundance of the major isotopologue and $R_{sum}-1 \approx 1-x_{626}$ is that fraction of the sample that is made up of all the minor isotopologues. Equivalently from Eq.
**Error! Reference source not found.** and the following paragraph it can be seen that $R_{sum}$ is the ratio of the total amount of $CO_2$ to that of the major isotopologue in a sample.

Abundances of the major and three singly-substituted isotopologues and $R_{sum}$ values for standard reference scales are given in **Error! Reference source not found.**. Abundances of the multiply-substituted isotopologues can be calculated following the examples of Eq.
**Error! Reference source not found.**. They are also listed for Hitran isotope ratios on the Hitran website https://www.cfa.harvard.edu/hitran/molecules.html.

*-Page 7, line 35 and following: IRMSs can also be used to measure absolute abundances of the single isotopologues, but the measurement is in general noisy and very difficult to calibrate. This "absolute" approach has been used in the beginning of mass spectrometry before it has been realized that a relative measurement with IRMS is much more precise than an absolute measurement[3]. This is also the case for isotopologue-specific optical systems. The noise in a line ratio is much smaller than the noise of the individual lines because there is a lot of correlated/technical noise on two simultaneously measured lines which cancels out in a ratio (given that the instrument can measure the targeted lines simultaneously in real-time as possible e.g. with a direct absorption spectrometer). It's not due to the nature of IRMS that the community has converged to go for relative δ-scales, but rather due to the more fundamental analytical advantage of measuring relative values with respect to a reference material as opposed to absolute measurements.*

This point has been addressed in the response to this referee's major comment on the absolute vs ratio calibration. Some sources of noise are common to the retrieval of amounts from two isotopologue absorption lines or bands, and these may ratio out, but fundamentally the spectrum is analysed for each isotopologue independently and then the results are ratioed. Technically this is also true of the ion counts in IRMS, but as the referee points out the absolute ion counts are very noisy, more so in IRMS than in retrievals from spectra.

---

## Author Comment (AC3) · 11 Oct 2018

**Author response to anonymous referee #2**

I thank referee 2 for his/her perceptive and constructive comments and make the following responses:

**General comment**

*My one general comment would be that although the approach taken by the author does indeed side-step the concentration dependence arising if one calibrates using a ratio-based approach, IRMS methods arose because of the precision advantage arising when measuring a ratio. It is certainly still the case that the precision with which any of the isotopologue-specific optical instruments can measure the minor isotopologues is limited 0.1 ppm (generously), so this will impact the precision with which one can determine a delta value from these instruments via this otherwise straightforward approach. As such, there remain advantages in working with the ratio even with instruments that 'naturally' lend themselves to this approach. Some acknowledgement of this ought to be made in the text.*

Please see detailed response to the same issue raised by referee 1, I have addressed both together.

**Specific comments**

*P 3, line 12: Clumped isotopes are mentioned, but not described, but as I see great value in this manuscript for students, I think it would be helpful to extend the parenthese to something like: (i.e. clumped isotopes, is the term for isotopologues carrying two (or more) of the heavy, rare isotopes.)*

Please see detailed response to referee 1 for the same issue.

*p 9, line11: 2018 should read 2017.*

Corrected.

*p 9. line 25: I think case 7 needs a little more explanation to be clear to readers.*

I have expanded case 7 as follows:

1) is a hypothetical standard with the isotopic composition of VPDB-CO$_2$. Examples include typical clean air (case 2), synthetic air synthesised with $^{13}$C-depleted CO$_2$ with $\delta^{13}$C = -35‰ (case 3), realistic errors of 2‰ in $\delta^{18}$O and $\delta^{17}$O (cases 4,5), and using isotope ratios assumed by Hitran rather than VPDB-CO$_2$ (case 6). Case 7 simulates the result if only singly-substituted isotopologues are included in the sum and all doubly-substituted minor isotopologues are ignored. Other cases can be assessed

---

## Author Response (AR2)

Dear Prof Fisher,

Thank you for your final comments on this manuscript, which I am happy to accommodate.

To address the issue of the precision of optical measurements I have strengthened the point in section 4.3 and the summary by breaking it out and expanding slightly into a separate paragraph in each case:

Section 4.3:

Optical FTIR and laser methods do not currently meet GAW requirements for repeatability of $\delta^{13}$C in $CO_2$ in clean background air measurements of 0.01‰ (WMO-GAW, 2016). Their precision is limited by the inherent signal:noise ratio of the optical measurement, not by the choice of absolute or ratio calibration. The precision currently available from optical measurements is nevertheless very useful for continuous analysis of air in non-baseline scenarios such as urban air or agricultural flux measurements.

Section 9:

Optical FTIR and laser methods do not currently meet GAW requirements for repeatability of $\delta^{13}$C in $CO_2$ in clean background air measurements (0.01‰). Their precision is currently limited by the inherent signal:noise ratio of the optical measurement, not by the calibration methodology. The precision currently available from optical measurements is nevertheless very useful for continuous analysis of air in non-baseline scenarios such as urban air or agricultural flux measurements.

I would prefer not to add this to the abstract – the paper is primarily concerned with the mechanism of the calibration procedures, not the actual precision of the measurements. The actual precision assessment is worth making in the text in the context of sample data, but I fear it would distract from the main point if added to the abstract.

I have made the minor corrections suggested in the re-revised MS. At page 10 line 17 (now line 22), I replaced "realistic" with "systematic".

The re-revised manuscript is appended below, with highlighted changes, and uploaded separately without highlighting.

Best wishes

David Griffith

[revised manuscript text omitted]
 | $\gamma_{626,meas}$ ppm | $\gamma_{636,meas}$ ppm | $\gamma_{628,meas}$ ppm | $\gamma_{626,cal}$ ppm | $\gamma_{636,cal}$ ppm | $\gamma_{628,cal}$ ppm | $^{13}r$ | $^{18}r$ | $R_{sum}$ | $\gamma_{CO2}$ ppm | $\delta^{13}C$ ‰ | $\delta^{18}O$ ‰ |
|---|---|---|---|---|---|---|---|---|---|---|---|---|
| 18:00 | 433.79 | 4.9845 | 1.9325 | 397.07 | 4.4015 | 1.6614 | 0.011085 | 0.002092 | 1.016117 | 403.47 | -8.53 | 1.76 |
| 00:00 | 492.97 | 5.6550 | 2.2211 | 450.80 | 4.9867 | 1.8891 | 0.011062 | 0.002095 | 1.016102 | 458.05 | -10.56 | 3.31 |
| 06:00 | 541.01 | 6.2000 | 2.4531 | 494.41 | 5.4624 | 2.0722 | 0.011048 | 0.002096 | 1.016088 | 502.37 | -11.80 | 3.46 |
| 12:00 | 433.37 | 4.9800 | 1.9309 | 396.69 | 4.3975 | 1.6601 | 0.011086 | 0.002092 | 1.016119 | 403.08 | -8.47 | 1.97 |

**Table 5. Actual isotopologue amounts and $R_{sum}$ values in 400 ppm total CO₂ for various isotopic compositions. The last column lists errors in calculating total CO₂ if different isotopic composition between reference (calibration) and sample measurements are not accounted for. See text for details of the various cases.**

| Case | $y_{CO2}$ ppm | $\delta^{13}C$ ‰ | $\delta^{18}O$ ‰ | $\delta^{17}O$ ‰ | $R_{sum}$ | $y_{626}$ ppm | $y_{636}$ ppm | $y_{628}$ ppm | $y_{CO2}$ error ppm |
|---|---|---|---|---|---|---|---|---|---|
| 1 | 400 | 0 | 0 | 0 | 1.01620 | 393.62 | 4.4077 | 1.6440 | 0.000 |
| 2 | 400 | −8 | 0 | 0 | 1.01611 | 393.66 | 4.3660 | 1.6442 | 0.035 |
| 3 | 400 | −35 | 0 | 0 | 1.01581 | 393.77 | 4.2484 | 1.6447 | 0.155 |
| 4 | 400 | 0 | 2 | 0 | 1.01621 | 393.62 | 4.4007 | 1.6473 | −0.003 |
| 5 | 400 | 0 | 0 | 2 | 1.01621 | 393.62 | 4.4007 | 1.6440 | −0.001 |
| 6 | 400 | 5.13 | −39.82 | −51.4 | 1.01605 | 393.68 | 4.4240 | 1.5788 | 0.060 |
| 7 | 400 | 0 | 0 | 0 | 1.01614 | 393.65 | 4.4010 | 1.6441 | 0.024 |

**Table 6. Details of isotopologues of common atmospheric species.**

| Species | Stable isotopes | No. isotopocules Total (distinct) | No. independent quantities to specify isotopic composition | $R_{sum}$ |
|---------|-----------------|-----------------------------------|-----------------------------------------------------------|-----------|
| $CO_2$ | $^{12}C$, $^{13}C$ $^{16}O$, $^{17}O$, $^{18}O$ | 18 (12) | 4 | $(1+^{13}r).(1+^{17}r+^{18}r)^2$ |
| $CH_4$ | $^{12}C$, $^{13}C$ $^{1}H$, $^{2}H$ | 32 (10) | 3 | $(1+^{13}r).(1+^{2}r)^4$ |
| $N_2O$ | $^{14}N$, $^{15}N$ $^{16}O$, $^{17}O$, $^{18}O$ | 12 (12) | 4 | $(1+^{15}r)^2.(1+^{17}r+^{18}r)$ |
| $CO$ | $^{12}C$, $^{13}C$ $^{16}O$, $^{17}O$, $^{18}O$ | 6 (6) | 4 | $(1+^{13}r).(1+^{17}r+^{18}r)$ |
| $H_2O$ | $^{1}H$, $^{2}H$ $^{16}O$, $^{17}O$, $^{18}O$ | 12 (9) | 4 | $(1+^{2}r)^2.(1+^{17}r+^{18}r)$ |

**Figures and Figure captions**

[Figure]

**Figure 1. Example of $\delta^{13}C$ dependence on $CO_2$ mole fraction for a Spectronus FTIR analyser. The measured data are fitted with a function of form of Eq. (15) with fitted parameters $\beta$ = - 1227 ‰ ppm and $\gamma$ = 0.0054 ‰ ppm$^{-1}$.**

[Figure]

**Figure 2. Calibration plots for three CO$_2$ isotopologues.**

[Figure]

**Figure 3. Calibrated total CO₂, δ¹³C and δ¹⁸O of sampled air on 23-24 Jan 2018 at a rural site in SE Australia. Air was sampled continuously, the displayed data are 1 minute averages.**

[Figure]

**Figure 4. Keeling plot of data shown in Figure 2.**